# Investigation of Quality of Life of Patients with Atopic Dermatitis and Quality of Life, Psychiatric Symptomatology, and Caregiver Burden of Their Mothers

**DOI:** 10.3390/children10091487

**Published:** 2023-08-31

**Authors:** Nülüfer Kilic, Mehmet Kilic

**Affiliations:** 1Department of Psychiatry, Elazığ Fethi Sekin City Hospital, Elazig 23280, Turkey; 2Division of Allergy and Immunology, Department of Pediatrics, Faculty of Medicine, University of Firat, Elazig 23119, Turkey; drmkilic@gmail.com

**Keywords:** atopic dermatitis, quality of life, depression, anxiety, caregiving burden, temperament characteristics

## Abstract

Background: The purpose of the present research was to evaluate the quality of life of patients with atopic dermatitis (AD), and that of their mothers. We compared the anxiety and depression scores and caregiving burden of mothers of patients with AD with the same scores of mothers of healthy children. Materials and Methods: A total of 153 patients between the ages of 2 months and 16 years with AD in our clinic, and their mothers, were contained in the patient group. An additional 141 healthy cases between the ages of 2 months and 16 years, and their mothers, were included as the control group. The Children’s Dermatology Life Quality Index (CDLQI) was completed according to the children’s opinions, and the Infant’s Dermatitis Quality of Life Index (IDLQI), Family Dermatology Life Quality Index (FDLQI), Beck Depression Inventory (BDI), Beck Anxiety Inventory (BAI), Temperament Evaluation of Memphis, Pisa, Paris and San Diego Autoquestionnaire, and Zarit Caregiver Burden (ZCB) scale were completed based on the mothers’ opinions. Results: We detected a positive relationship between the SCORAD index and the IDLQI, CDLQI, and FDLQI scores of children with AD. We found that anxiety, depression, and caregiving burden in mothers of AD cases increased when mothers of AD cases were compared with mothers of healthy children (*p* < 0.0001, *p* < 0.0001, and *p* < 0.002, respectively). Also, based on the mothers’ responses, we noted a positive correlation among the BDI, BAI, ZCB, and SCORAD index scores. Conclusions: Our study found that the quality of life of patients with AD, and that of their mothers, was negatively affected by the disease. We also found that anxiety, depression levels (not at the clinical diagnosis level, and the caregiving burden in mothers of AD cases increased when mothers of AD cases were compared with mothers of healthy children.

## 1. Introduction

Atopic dermatitis (AD) is a recurrent, itchy, and inflammatory skin disease that is common in children, with an incidence rate varying between 8.7% and 18%. The prevalence of AD has been increasing in recent years [1]. AD is characterized by periods of exacerbation and recovery. Although the factors that are most responsible for exacerbations in childhood are foods such as cow’s milk and eggs, aeroallergens and pollen-related foods come to the forefront among adult AD patients. The frequency of food allergy in patients with AD has been described in different studies as being 33–63%. Asthma may also develop in more than 50% of patients with AD, while allergic rhinitis manifests in approximately 75% of them [2]. 

Atopic dermatitis in children is divided into three groups according to age. Infants (≤2 years): In this period vesicular lesions, serous exudates, and crusts in severe cases are observed. During this period, facial and extensor region involvement is typical, but any region can be affected. Lichenification is not expected during this period. Childhood (2–12 years): In this period, eczematous lesions develop in flexural areas (antecubital fossa, neck, hand, ankles). Lesions show lichenification rather than exudation. Recurrent lesions on the neck increase pigmentation (dirty neck). In adolescents and adults (13 years and older): In this period, eczema lesions are more localized, and chronic eczematous lesions are detected. Lesions are mostly seen in the hand, flexural region, and head and neck region [3,4]. In the treatment of atopic dermatitis, topical emollients to moisturize the skin, corticosteroids to relieve inflammation, and antibiotics to prevent staphylococcal colonization are generally used. As an alternative treatment, topical calcineurin inhibitors, cyclosporine, omalizumab, and dupilumab treatments can be used [3].

Atopic dermatitis impairs quality of life of children. In addition, AD negatively affects the quality of life of their families, physically, psychologically, and psychosocially. It causes itching, pain, insomnia, fatigue, and mood changes in children. The family members’ quality of life is influenced negatively by factors such as having a child with AD; feelings of stress, sadness, anger, embarrassment, and guilt; depression; anxiety; restrictions in social activities; and decreased work performance [5,6,7,8,9,10,11,12,13]. Furthermore, it often takes a lot of time to administer treatments for a child with eczema. Parents, and especially mothers who are also their child’s primary caregiver, must manage complex skin treatments. This also leads to a greater care burden on the mother because of the dependence of the child with AD on their primary caregiver [5,10,12,14,15,16,17,18,19,20]. 

We assessed the quality of life of patients diagnosed with AD, and their mothers. We also aimed to compare the anxiety and depression scores and caregiving burden of mothers of patients with AD with those of mothers of healthy cases.

## 2. Materials and Methods

A total of 153 patients between the ages of 2 months and 16 years who had been diagnosed with AD at our pediatric allergy and immunology clinic, and their mothers, were contained in the study group. An additional 141 healthy children between the ages of 2 months and 16 years, and their mothers, were included as the control group. Those with any syndromic disease; malignancy; metabolic disease; immunodeficiency; chronic heart disease; diseases of the endocrine, kidney, lung, gastrointestinal, liver, and/or central nervous systems; and individuals with a family history of other chronic diseases were eliminated from both groups. Children with an exacerbation of lesions in the 14 days prior to the start of the study were also excluded from the study group. All children’s AD diagnoses were made according to the Hanifin–Rajka criteria, and they were enrolled in the study group [21].

The Scoring Atopic Dermatitis (SCORAD) and objective SCORAD (SCORAD-O) indices were used to evaluate the severity of AD. A SCORAD score can be rated between 0 and 103, while SCORAD-O scores range from 0 to 83. According to the SCORAD evaluation, those rated 0–24 were described as having mild AD, those with scores 25–50 were deemed to have moderate AD, and those with scores 51–103 were said to have severe AD. Patients’ subjective symptoms (i.e., itching and sleepiness) were not assessed in the SCORAD-O index, but were divided into three groups: Those between 0–15 were classified as mild, those between 15–40 were classified as moderate, and those >40 scores were classified as severe [22]. Participants in the patient group were skin prick-tested with foods and/or inhaled allergens, and allergen-specific immunoglobulin E (IgE) levels were evaluated. The children in this group were diagnosed with an IgE-mediated food allergy based on a positive skin test and/or positive food challenge test with a positive specific IgE, a convincing history of a severe allergic reaction to food sensitization according to current guidelines (based on high predictive power), or a history of anaphylaxis in the last six months [23]. The research was authorized by the Ethics Committee of Fırat University, and a written informed permission document was got from mothers and fathers of each child included in the study.

### 2.1. Questionnaires

#### 2.1.1. The Infants’ Dermatitis Quality of Life Index

The Infants’ Dermatitis Quality of Life index (IDQOL) [24] enables a parent or caregiving representative to evaluate the quality of life of a child with AD under the age of 4 using a 10-item questionnaire that covers the activities of the past week. This questionnaire also measures the perceived impact of itching, mood, sleep time, play or swimming, daily activities, eating, medical therapy, dressing and undressing, and taking a shower on quality of life. There is also an additional question evaluating the severity of eczema by parent/caregiver (IDLQI1). Each question in the questionnaire is scored between 0 and 3 and the highest total value is 30. Higher scores indicate a deterioration in quality of life. The Dermatological Quality of Life index (IDLQI2), for cases between aged 2 months to 6 years, was completed by the mothers of children in this study.

#### 2.1.2. The Children’s Dermatology Life Quality Index

The Children’s Dermatology Life Quality Index (CDLQI) [25] evaluates the quality of life in patients diagnosed with AD who are between the ages of 5 and 16 years. It contains 10 questions covering the past week’s activities, measuring the perceived impact of symptoms; emotions; leisure, school and vacation activities; personal relationships; sleep; and medical treatments on their quality of life. Each question in the questionnaire is scored between 0 and 3 and the highest total value is 30. Higher scores indicate a greater deterioration in quality of life. The CDLQI was completed by the children in the study who have been diagnosed with AD.

The original recommendation regarding these indices, as described by Finlay et al. [24,25], was to assess the quality of life in cases who diagnosed AD up to age 4 using the IDQOL and the CDLQI for those 5 to 16 years old. These age recommendations may be appropriate for children in the United Kingdom (UK), because children in the UK start school at the age of 4. In this context, it is assumed that they can read and complete a questionnaire on their own once they have reached the age of 5. However, in Turkey, children start school at the age of 6. We suggest that 7 is the earliest age at which a child could complete the questionnaire on their own, or with only a little help from a parent. For this reason, we used the IDQOL for cases between the ages of 2 months and 6 years and the CDLQ for children ages 7–16.

#### 2.1.3. Family Dermatology Life Quality Index

The Family Dermatology Life Quality index (FDLQI) evaluates the effects of any dermatological diseases of cases of all ages and the quality of life of adults in the family [26]. It is a 10-item questionnaire covering the activities of the 30 days. Each question in the questionnaire is scored between 0 and 3. Questions address the areas of affective and physical condition, relationships, leisure pursuits, daily life, care burden, impacts on work/education, housework, and expenditures. The higher the score on the questionnaire, the more deteriorated the quality of life. The mothers of all children participating in the study competed the FDLQI.

#### 2.1.4. Beck Depression Inventory

Beck et al. [27] developed the Beck Depression Inventory (BDI) to evaluate the affective, cognitive, and motivational symptoms of depressive disorder. The cut-off point of the scale, for which Turkish validity and reliability studies have been conducted, was accepted as 17. The scale contains 21 questions, each of which is scored between 0 and 3. The scores obtained from the scale are assessed as normal, for scores of 1–10; moderate mood disorder, for scores 11–16; clinical depression, for scores 17–20; moderate depression, for scores 21–30; serious depression for scores 31–40; and severe depression for scores 41–63.

#### 2.1.5. Beck Anxiety Inventory

Beck et al. [28] developed the Beck Anxiety Inventory (BAI) in 1988 to evaluate anxiety symptoms experienced by individuals. It has been demonstrated for validity and reliability in the Turkish context. This scale contains 21 questions, and each question is scored between 0 and 3. When the scores obtained from the scale are assessed, those between 0–7 were classified as low anxiety, those between 8–15 were classified as mild anxiety, those between 16–25 were classified as moderate anxiety, and those between 26–63 were classified as a high level of anxiety.

#### 2.1.6. Temperament Evaluation of Memphis, Pisa, Paris, San Diego Autoquestionnaire

The Temperament Evaluation of Memphis, Pisa, Paris, San Diego Autoquestionnaire was used by Akiskal et al. [29] to assess the dominant emotional temperament in individuals. The validity and reliability study of the scale was conducted on patients in Turkey. Turkish study includes depressive, hyperthymic, irritable, cyclothymic, and anxious temperaments. It can be said that in which temperament the scores are above the cut-off point, that person has that temperament. When more than one temperament cut-off point is exceeded, the presence of more than one dominant temperament is considered.

#### 2.1.7. Zarit’s Caregiver Burden Scale 

Zarit et al. [30] developed the Caregiver Burden Scale (ZCB) in 1980 to measure stress in caregivers of people with care needs. The ZCB has been validated and deemed reliable for the Turkish context. The scale includes mental tension and the decreased personal quality of life, limitations and restrictions, difficulty in daily activities, economic burdens, and dependency. The scoring range is between 22 and 110. Scores from 22–46 are considered to reveal a mild burden, those between 47–55 imply a moderate burden, and those 56–110 indicate a severe burden.

### 2.2. Statistical Analysis

The data obtained in the study were assessed statistically using Statistical Package for Social Sciences, v. 22 software. (The data in our study were assessed statistically using Statistical Package for Social Sciences, v. 22 software.) Whether the data were normally distributed or not was assessed with the Kolmogorov–Smirnov test. The *t*-test was used for independent samples to compare the means of the two groups for continuous variables with normal distribution, and descriptive values were written as statistical mean ± standard deviation. The Mann–Whitney U test was used to make median comparisons between the groups for continuous variables that did not show normal distribution, while the descriptive statistics were given as median and minimum–maximum. Pearson’s correlation coefficient was applied for the relationship between continuous variables (which are normally distributed). The Spearman’s rho correlation coefficient was used for the relationship between continuous variables (which are non-normally distributed). *p* values < 0.05 were considered significant.

## 3. Results

A total of 161 patients with a diagnosis of AD, and their mothers, were contained in the patient group. A group of 147 healthy children and their mothers served as the control group. Ultimately, 153 mothers in the patient group and 141 mothers in the control group were able to complete the study. The average age of onset of the complaints in the children with AD was 6 (2–54) months, the duration of the disease was 11 (1–178) months, and the clinical follow-up period was 8 (1–178) months in the study group (Table 1). There were 87 boys (56.9%) in the study group, and their mean age was 18 (3–192) months. There were 74 boys (52.5%) in the control group, and their mean age was 18 (6–132) months. No differences were detected between the study and control groups, in terms of mean age and gender of cases (*p* > 0.05). The mean age of mothers in the study group was 28 (19–50), as was the mean age of mothers in the control group (20–40). No statistical relationship was detected between two groups, with regard to the number of children in the family, number of people in the household, household income, mother’s marital data, and mother’s chronic disease status (*p* > 0.05). We did note a statistical difference in terms of mothers’ employment status and education levels (*p* < 0.05) (Table 2). 

When the BDI and BAI scores of the mothers of children with eczema were compared with the mothers of healthy children; mothers of children with eczema had higher scores (*p* < 0.0001, *p* < 0.0001, respectively). Also, we found higher expressive, cyclothymic, and anxious temperament scores in mothers of children with eczema when compared to mothers of healthy children (*p* = 0.0001, *p* = 0.047, *p* = 0.0001, respectively). Mothers of children with AD had a higher caregiver burden (CB), compared to mothers of healthy children (*p* = 0.002) (Table 3). 

According to the reports by mothers of children with AD, we detected a positive correlation between BDI and the SCORAD index, SCORAD objective index, IDLQI, CDLQI, and FDLQI. Similarly, we noted a positive relationship between the CB of mothers and the SCORAD index, SCORAD objective index, IDLQI, and FDLQI. We also observed a positive correlation between the BAI and the SCORAD index, SCORAD objective index, depressive, cyclothymic temperaments, IDLQI, and FDLQI. No correlation was detected between mothers’ education levels, occupations, family income levels, and BDI and BAI scores (Table 4).

According to the mothers’ reports, we found a positive relationship between FDLQI and the SCORAD index, SCORAD objective index, ZCB, IDLQI, and CDLQI. In addition, we observed a positive relationship between IDLQI and the SCORAD index, SCORAD objective index and FDLQI. According to the reports from children with AD, we detected a positive relationship between CDLQI and the age of AD symptoms’ onset, SCORAD index, SCORAD-Objective index, ZCB, and FDLQI. We also found a positive relationship between the detection of food allergy in patients with AD and BAI, CDLQI, and FDLQI (Table 4).

## 4. Discussion

Unlike any other organ disease, AD is generally not mortal, but it can affect the person’s appearance and cause negative effects, with regard to interpersonal relationship, social and routine activities. Previous studies have reported that AD reduces patient’s quality of life [6,13,15,25,31]. Van Oosterhout et al. [32] assessed the IDQOL and CDLQI levels of patients with AD. The mean CDLQI result they found was significantly higher in children in the ages of 4 and 16 years, compared to children aged 0–4 years. In another study, conducted in the Czech Republic, cases with AD were classified into three groups (0–6 years, 7–13 years, and 14–18 years) and their CDLQI and IDQOL scores were gathered. No significant differences were detected among the quality-of-life indices for these three age groups [33]. Similarly, in a study that investigated the quality of life of patients with AD aged 0–16, patients were divided into three groups, cases between 0–1 aged, cases between 1–6 aged, and cases in >6 years. No significant differences were detected among these three groups, with regard to quality-of-life index scores [34]. We determined no statistical differences among age groups, in terms of and IDQOL and CDLQI scores, in our study. It has been reported that, as the severity of eczema increases in AD cases, children’s quality of life deteriorates significantly [31]. We also determined that children’s quality of life scores increased (meaning, the quality declined) as the severity of eczema increased, and there was a positive relationship observed between CDLQI score and food allergies in AD cases. We think that the quality of life of patients with AD deteriorates further because of additional diagnostic testing, including oral food provocation tests and the uncomfortable dietary restrictions associated with concomitant food allergies. 

Previous studies have reported that having a child with AD has a significant influence on a family’s quality of life [10,14,16,20,35]. Studies report that the quality of life of parents of AD cases is reduced compared to parents with healthy children [16,20]. In a study evaluating the quality of life of parents of AD cases, only 3.4% reported that the quality of life was not affected. By contrast, 23.3% were mildly influenced, 66.4% were moderately influenced, and 6.9% were severely influenced. The same study reported that quality of life deteriorated significantly as the seriousness of atopic dermatitis increased [14]. In our study, we also found that, as the severity of eczema increased, the family’s quality-of-life scores increased (meaning their quality of life declined), and we detected a positive correlation between concomitant food allergies and FDLQI in children with AD. These results show that the quality of life of parents, and especially mothers, who care for children with eczema is significantly influenced by the disease. We believe that the quality of life of patients, and their families, might be assessed during outpatient following and necessary measures must be taken to increase the quality of life.

In previous studies, it has been reported that anxiety and/or depression levels increase in families caring for eczema patients [7,11,36]. A study conducted in Australia reported that having a patient with moderate or severe eczema causes more anxiety than caring for a child with Type I Diabetes [18]. In a study conducted with parents of children with AD, it was reported that parents experienced more sleeplessness, anxiety, and depressed mood, compared to families of children with asthma [37]. In another study, from China, 41.5% of parents of patients with eczema defined symptoms of anxiety, 39.6% had symptoms of depression, and 29.7% had both [8]. Similarly, 36% of parents who cared for patients with eczema had manifestations of anxiety and 36% had depression. However, no statistical correlation was detected between the quality of life of patients with eczema and anxiety and depression in parents [38]. It has also been published that mothers of patients with eczema feel more guilt about their children’s symptoms, and their parental stress scores were higher, compared to fathers [9,39]. In our study, we found that mothers of children with eczema had significantly increased anxiety and depression levels, compared to mothers of healthy children. High maternal levels of anxiety symptoms can be explained by factors such as uncertainty regarding the prognosis of the illness; difficulties of medical care, such as daily topical applications; uncomfortable dietary restrictions in children with food allergies; and the fact that AD is a disease characterized by relapses. We also think that guilt about passing a genetic predisposition along to the patient, sadness due to the patient’s physical appearance, and the fact that AD is not a disease that can be cured in a short time cause depression in mothers. 

It has been reported that parents’ stress and depression symptoms increase as children’s AD severity increases [8,40]. Su et al. [8] declared that families of patients with high AD severity scores had higher anxiety symptoms. Similarly, our results revealed a positive correlation between the mothers’ BDI and BAI scores. These data confirm that maternal anxiety and depression symptoms increase as eczema severity increases in patients with eczema. We also observed a significant relationship between the detection of food allergies in patients with eczema and BAI. We think that the mothers’ anxiety scores increase because of additional diagnostic tests, including oral food provocation and uncomfortable dietary restrictions for food allergies, which accompany AD. Further, in our research, when the mothers of patients) with food allergies were compared with the mothers in the control group, we found that depressive, anxious, and cyclothymic temperament characteristics were higher. Temperament characteristics are the basis of mood disorders, and individuals’ dominant temperamental characteristics cause a predisposition to mood disorders [41]. We believe that certain temperamental characteristics in the mothers of patients with eczema can be used as a precursor for psychological reactions to the child’s disease and for predicting comorbid psychiatric diseases that may develop later.

Caring for and treating a child with AD often requires a lot of time. Parents, and especially mothers, must manage complex skin treatments, which are usually seen by sleeplessness, feelings of disappointment, despondency, unable to deal, lifestyle changes, restricted social activities, and financial burdens [10,12,14,15,16,17,18,19,20]. Mothers of patients with eczema show their affection more through hygienic rituals, which transforms the relationship between mother and child into one of a nurse–patient relationship [19]. Lawson et al. reported that 90% of families with patients with eczema experienced difficulties related to care (e.g., increased washing and cleaning load, difficulties in choosing food, and shopping difficulties) [16]. Su et al. [18] estimated that 2–3 h per day were strived caring for a patient with eczema. In this study, we found that the care burden of mothers of patients with eczema was higher than that of the control group. We also observed a positive correlation between mothers’ caregiving burden and children’s eczema severity scores. In light of these data, we think it would be useful to apply measurement tools that define caregivers and reveal support factors to prevent early burnout due to the caregiving role.

### Strengths and Limitations

The study has case and control groups, the diversity of the number of scales used, and the validity and reliability of these scales in the Turkish context are the strengths of the study. However, our study had some limitations. First, our study is a cross-sectional design. Second, only anxiety and depression scores of mothers were evaluated in the study, mothers were not followed up for these disorders during long time. Third, the inability to use structured interview techniques; the existence of individual perceptions and differences; and the lack of data, such as parents’ knowledge of the disease, hampered the study. Finally, the lack of objective burden variables (i.e., distance from the treatment center, social support for the caregiver, the presence of assistant caregivers, sleep pattern and quality) can be considered another limitation.

## 5. Conclusions

This study found that the quality of life of patients with eczema, and that of their mothers, is negatively affected by AD. We also found that mothers of patients with eczema have higher anxiety and depression levels, and higher caregiving burdens, than mothers in the control group, although not at the clinical diagnosis level. It is important to determine the quality of life, psychological symptoms, and caregiving burden of parents, and especially mothers, for the compliance and success of AD treatment. Based on these results, we believe that multidisciplinary cooperation must be established with the relevant departments to take preventive measures in this respect.

## Figures and Tables

**Table 1 children-10-01487-t001:** The data of patients with atopic dermatitis.

Parameters	Data
Age of AD symptoms onset (month)	6 (2–54) *
Duration of AD (month)	11 (1–178)
Duration of follow-up of AD (month)	8 (1–178)
SCORAD index	26.5 (7.2–91)
SCORAD classification *n* (%)	
Mild	67 (43.8)
Moderate	57 (37.3)
Severe	29 (19)
SCORAD-objective index	17.6 (3.7–73)
SCORAD-objective classification *n* (%)	
Mild	61 (39.9)
Moderate	65 (42.5)
Severe	27 (17.6)
IDLQI1	2 (1–4)
IDLQI2	9 (2–27)
CDLQI	9 (4–24)
FDLQI	10 (2–25)
Food allergen sensitivity *n* (%)	
Yes	71 (46.4)
No	82 (53.6)
Concomitant other allergic disease *n* (%) ^†^	
Yes	74 (48.4)
No	79 (51.6)

* = median (minimum–maximum); ^†^ = Asthma and/or allergic rhinitis; SCORAD = Scoring Atopic Dermatitis; AD = Atopic dermatitis; IDLQI = Infants’ Dermatology Life Quality index; CDLQI = Childrens’ Dermatology Life Quality index; FDLQI = Family Dermatology Life Quality index.

**Table 2 children-10-01487-t002:** The sociodemographic data of all groups.

Parameters	Atopic Dermatitis Children (*n* = 153)	Healthy Children (*n* = 141)	*p* Value
Children’s gender *n* (%)			0.483
Boy	87 (56.9)	74 (52.5)	
Girl	66 (43.1)	67 (47.5)	
Children’s age (month)	18 (3–192) *	18 (6–132)	0.759
Children’s age groups (month) *n* (%)			0.375
4–72	137 (89.5)	121 (85.8)	
73–192	16 (10.5)	20 (14.2)	
Mother’s age (year)	28 (19–50)	28 (20–40)	0.622
No. of children within family *n* (%)	2 (1–6)	2 (1–5)	0.088
No. of person in household *n* (%)	4 (2–10)	4 (3–7)	0.057
Mother’s marital status *n* (%)			0.847
Married	139 (90.8)	129 (91.5)	
Divorced	14 (9.2)	12 (8.5)	
Mother’s chronic disease *n* (%)			0.558
Yes	17 (11.1)	12 (8.5)	
No	136 (88.9)	129 (91.5)	
Mother’s psychiatric treatment *n* (%)			0.854
Yes	18 (11.8)	15 (10.6)	
No	135 (88.2)	126 (89.4)	
Mother’s academic level *n* (%)			0.001
Elementary school	29 (19)	35 (24.8)	
İntermediate school	28 (18.3)	7 (5)	
High school	54 (35.3)	41 (29.1)	
University	42 (27.5)	58 (41.1)	
Mother’s employment status *n* (%)			0.010
Housewife	98 (64.1)	66 (46.8)	
Self-employment	21 (13.7)	33 (23.4)	
Officer	34 (22.2)	42 (29.8)	
Family income *n* (%)			0.403
<300 $	7 (4.6)	9 (6.4)	
300–450 $	16 (10.5)	16 (11.3)	
450–650 $	65 (42.5)	41 (29.1)	
650–750 $	30 (19.6)	26 (18.4)	
>750 $	35 (22.9)	49 (34.8)	

* = Median (minimum–maximum).

**Table 3 children-10-01487-t003:** Evaluation of depression, anxiety, temperament types, and caregiving burdens of mothers in all groups.

Parameters	Atopic Dermatitis Children (*n* = 168)	Healthy Children (*n* = 152)	*p* Value
Depressive temperament	6 (1–20) *	5 (0–17)	0.002
Cyclothymic temperament	5 (0–16)	5 (0–16)	0.047
Hyperthymic temperament	8 (0–18)	9 (0–19)	0.068
Irritable temperament	2 (0–10)	1 (0–11)	0.180
Anxious temperament	6 (0–23)	5 (0–22)	0.0001
Beck depression inventory	8 (0–38)	4 (0–21)	0.0001
Beck anxiety inventory	8 (0–51)	4 (0–41)	0.0001
Zarit caregiver burden scale	44 (24–71)	39 (22–66)	0.002

* = Median (minimum–maximum).

**Table 4 children-10-01487-t004:** Correlation between mother anxiety, depression, caregiver burden, and children’s atopic dermatitis quality of life in terms of the opinions of mothers in the study group.

Characteristics	BDI	BAI	ZCBS	FDLQI	IDLQI	CDLQI
Children’s age	r = 0.010 *p* = 0.902	r = −0.056 *p* = 0.491	r = −0.042 *p* = 0.606	r = −0.018 *p* = 0.829	r = −0.154 *p* = 0.074	r = 0.194 *p* = 0.455
Mother’s age	r = 0.051 *p* = 0.531	r = −0.025 *p* = 0.759	r = −0.060 *p* = 0.464	r = −0.052 *p* = 0.527	r = −0.119 *p* = 0.169	r = 0.363 *p* = 0.152
The age of AD symptoms onset	r = 0.011 *p* = 0.891	r = −0.075 *p* = 0.355	r = −0.005 *p* = 0.948	r = 0.019 *p* = 0.814	r = −0.084 *p* = 0.330	r = 0.486 *p* = 0.048
Duration of AD	r = 0.018 *p* = 0.822	r = −0.043 *p* = 0.595	r = −0.084 *p* = 0.303	r = −0.020 *p* = 0.809	r = −0.140 *p* = 0.105	r = 0.108 *p* = 0.680
Duration of follow-up of AD	r = 0.032 *p* = 0.693	r = −0.039 *p* = 0.635	r = −0.045 *p* = 0.577	r = −0.019 *p* = 0.813	r = −0.144 *p* = 0.094	r = 0.135 *p* = 0.604
SCORAD index	r = 0.254 *p* = 0.002	r = 0.303 *p* = 0.000	r = 0.271 *p* = 0.001	r = 0.605 *p* = 0.000	r = 0.646 *p* = 0.000	r = 0.649 *p* = 0.005
SCORAD—objective index	r = 0.260 *p* = 0.001	r = 0.237 *p* = 0.000	r = 0.256 *p* = 0.001	r = 0.598 *p* = 0.000	r = 0.639 *p* = 0.000	r = 0.552 *p* = 0.022
No. of children within family	r = 0.062 *p* = 0.444	r = −0.006 *p* = 0.946	r = −0.030 *p* = 0.714	r = −0.062 *p* = 0.466	r = −0.085 *p* = 0.324	r = 0.033 *p* = 0.901
No. of person in household	r = 0.055 *p* = 0.500	r = 0.036 *p* = 0.658	r = −0.060 *p* = 0.461	r = −0.059 *p* = 0.466	r = −0.068 *p* = 0.429	r = 0.104 *p* = 0.691
Depressive temperament	r = −0.077 *p* = 0.341	r = 0.104 *p* = 0.202	r = 0.075 *p* = 0.358	r = 0.089 *p* = 0.276	r = 0.026 *p* = 0.761	r = 0.053 *p* = 0.841
Cyclothymic temperament	r = −0.043 *p* = 0.601	r = 0.331 *p* = 0.000	r = 0.010 *p* = 0.904	r = 0.038 *p* = 0.276	r = 0.104 *p* = 0.227	r = 0.083 *p* = 0.751
Hyperthymic temperament	r = −0.061 *p* = 0.455	r = −0.196 *p* = 0.015	r = −0.135 *p* = 0.097	r = −0.146 *p* = 0.071	r = −0.109 *p* = 0.207	r = −0.622 *p* = 0.008
Irritable temperament	r = 0.031 *p* = 0.703	r = 0.136 *p* = 0.094	r = 0.049 *p* = 0.552	r = 0.000 *p* = 0.997	r = −0.017 *p* = 0.845	r = 0.315 *p* = 0.219
Anxious temperament	r = 0.027 *p* = 0.745	r = 0.167 *p* = 0.039	r = 0.059 *p* = 0.470	r = 0.112 *p* = 0.167	r = 0.173 *p* = 0.044	r = −0.051 *p* = 0.847
ZCBS	r = 0.070 *p* = 0.390	r = 0.013 *p* = 0.872	----	r = 0.178 *p* = 0.028	r = 0.379 *p* = 0.134	r = 0.194 *p* = 0.024
IDLQI	r = 0.282 *p* = 0.001	r = 0.235 *p* = 0.006	r = 0.194 *p* = 0.024	r = 0.877 *p* = 0.000	----	----
CDLQI	r = 0.487 *p* = 0.047	r = 0.400 *p* = 0.112	r = 0.379 *p* = 0.134	r = 0.783 *p* = 0.000	----	----
FDLQI	r = 0.227 *p* = 0.005	r = 0.205 *p* = 0.011	r = 0.178 *p* = 0.028	----	r = 0.877 *p* = 0.000	r = 0.783 *p* = 0.000

AD = Atopic dermatitis, SCORAD = Scoring Atopic Dermatitis, BDI = Beck Depression Inventory, BAI = Beck Anxiety Inventory, ZCBS = Zarit Caregiver Burden Scale, FDLQI = Family Dermatology Life Quality index, IDLQI = Infants’ Dermatology Life Quality index, CDLQI = Childrens’ Dermatology Life Quality index.

## Data Availability

Data are available upon reasonable request. The data that support the findings of this study are available on request from the corresponding author.

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
