# Peer review of "Investigation of Quality of Life of Patients with Atopic Dermatitis and Quality of Life, Psychiatric Symptomatology, and Caregiver Burden of Their Mothers"

_children, 2023, doi:10.3390/children10091487_

Round 1

Reviewer 1 Report

Original and well-written article

Good materials and methods , right sampling, results and conclusions well represented

I have only two minor revisions for the authors 

1) The introduction needs to be enriched , we need to talk about the treatments and phenotypes of ad in children, I leave a useful article for authors to use 

- DOI: 10.1111/dth.15901

2) Add paragraph limitations of the study 

3) minimal language review is required

Minor editing of English language required

Author Response

Reviwer 1

Original and well-written article

Good materials and methods, right sampling, results and conclusions well represented

I have only two minor revisions for the authors

Response to Reviewer 1

1) The introduction needs to be enriched , we need to talk about the treatments and phenotypes of ad in children,

I leave a useful article for authors to use

- DOI: 10.1111/dth.15901

Answer: Information about the treatments and phenotypes of atopic dermatitis in childhood was written in the introductory part of the article and a reference was given.

2) Add paragraph limitations of the study

Answer: We added paragraph limitations of the study

3) Minimal language review is required

Comments on the Quality of English Language Minor editing of English language required

Answer: English language of the manuscript were checked by Scribendi Editing and Proofreading Services.

Reviewer 2 Report

In this article, Kilic et al assessed the quality of life of children with AD and their mothers. They compared the anxiety and depression scores and caregiving burden of mothers of children with AD with the mothers of healthy children. In this study, 153 AD children and 141 healthy children their mothers were included. They reported that a positive correlation was detected between the SCORAD Index and IDLQI, CDLQI, and FDLQI of children with AD. They also found that mothers of children with AD had higher levels of anxiety, depression, and caregiving burden compared to the mothers in the control group. Also, according to the mothers’ opinions, a positive correlation was detected between BDI, BAI, ZCBS, and SCORAD Index. They concluded that the quality of life of children with AD and their mothers was affected negativel, and that mothers with children with AD had higher anxiety and depression scores and higher caregiving burden compared to the mothers in the control group, although not at the clinical diagnosis level. This is an excellent study focusing on children and families of AD patients. Although the results were expected, this is an excellent study that actually demonstrated the results with data. According to this study, children with AD need psychological care not only for the patients themselves but also for those who care for them. There are no particular major concerns, but there was one minor concern.

minor concern)

1) In table 4, the r and p values for SCORAD and SCORAD-o are different from other notations in the same table. Please unify the notation.

Author Response

Reviwer 2

Comments and Suggestions for Authors

In this article, Kilic et al assessed the quality of life of children with AD and their mothers. They compared the anxiety and depression scores and caregiving burden of mothers of children with AD with the mothers of healthy children. In this study, 153 AD children and 141 healthy children their mothers were included. They reported that a positive correlation was detected between the SCORAD Index and IDLQI, CDLQI, and FDLQI of children with AD. They also found that mothers of children with AD had higher levels of anxiety, depression, and caregiving burden compared to the mothers in the control group. Also, according to the mothers’ opinions, a positive correlation was detected between BDI, BAI, ZCBS, and SCORAD Index. They concluded that the quality of life of children with AD and their mothers was affected negativel, and that mothers with children with AD had higher anxiety and depression scores and higher caregiving burden compared to the mothers in the control group, although not at the clinical diagnosis level. This is an excellent study focusing on children and families of AD patients. Although the results were expected, this is an excellent study that actually demonstrated the results with data. According to this study, children with AD need psychological care not only for the patients themselves but also for those who care for them. There are no particular major concerns, but there was one minor concern.

minor concern)

Response to Reviewer 2

1) In table 4, the r and p values for SCORAD and SCORAD-o are different from other notations in the same table. Please unify the notation.

Answer: We wrote the r and p values for SCORAD and SCORAD-O in Table 4.
